# Stille coupling via C–N bond cleavage

Dong-Yu Wang[1,2], Masatoshi Kawahata[3], Ze-Kun Yang[1], Kazunori Miyamoto[1], Shinsuke Komagawa[4], Kentaro Yamaguchi[3], Chao Wang[1,2] & Masanobu Uchiyama[1,2]

Cross-coupling is a fundamental reaction in the synthesis of functional molecules, and has been widely applied, for example, to phenols, anilines, alcohols, amines and their derivatives. Here we report the Ni-catalysed Stille cross-coupling reaction of quaternary ammonium salts via C–N bond cleavage. Aryl/alkyl-trimethylammonium salts $[Ar/R–NMe_3]^+$ react smoothly with arylstannanes in 1:1 molar ratio in the presence of a catalytic amount of commercially available $Ni(cod)_2$ and imidazole ligand together with 3.0 equivalents of CsF, affording the corresponding biaryl with broad functional group compatibility. The reaction pathway, including C–N bond cleavage step, is proposed based on the experimental and computational findings, as well as isolation and single-crystal X-ray diffraction analysis of Ni-containing intermediates. This reaction should be widely applicable for transformation of amines/ quaternary ammonium salts into multi-aromatics.

[1] Graduate School of Pharmaceutical Sciences, University of Tokyo, 7-3-1 Hongo, Bunkyo-ku, Tokyo-to 113-0033, Japan. [2] Advanced Elements Chemistry Research Team, RIKEN Center for Sustainable Resource Science, and Elements Chemistry Laboratory, RIKEN, 2-1 Hirosawa, Wako-shi, Saitama-ken 351-0198, Japan. [3] Faculty of Pharmaceutical Sciences at Kagawa Campus, Tokushima Bunri University, 1314-1 Shido, Sanuki-shi, Kagawa-ken 769-2193, Japan. [4] Graduate School of Pharmaceutical Sciences, Osaka University, 1-6 Yamadaoka, Suita-shi, Osaka-fu 565-0871, Japan. Correspondence and requests for materials should be addressed to K.Y. (email: kyamaguchi@kph.bunri-u.ac.jp) or to C.W. (email: chaowang@mol.f.u-tokyo.ac.jp.) or to M.U. (email: uchiyama@mol.f.u-tokyo.ac.jp).

The Pd-catalysed Stille cross-coupling reaction between stannanes and organic halides (RX) was discovered during 1976–1978 (refs 1–3), and is currently the second most widely used cross-coupling method for the synthesis of functional molecules and polymers, both in laboratory research and in industry[4–10]. However, Stille coupling using C–O[11–14] and C–N[15–22] electrophiles, such as phenol and amine derivatives, has hardly been investigated[23–25], even though such compounds are frequently encountered in synthetic investigations.

Amine groups occur widely in natural products, and are also found in many pharmaceuticals, dyes and functional molecules. A large variety of amines are commercially available, mostly at reasonable cost. On the other hand, transformation of the $NR_2$ group is generally difficult, due to the chemical inertness of the C − N bond. Therefore, efficient C − N bond conversion methods suitable for late-stage functionalization would greatly expand the utility of amine compounds as synthetic feedstocks. Quaternary organo-ammonium salts can be easily prepared from various aryl/alkyl amines. Their potential usage in cross-coupling was pioneered by Wenkert et al.[26] (Ni-catalyst and Grignard reagent). Other protocols have been developed by MacMillan[27] (Ni-catalyst and arylboron reagent), Wang[28] (Ni-catalyst and organozinc reagent), Watson[29] (Ni-catalyst and aryl/alkylboron reagent) and others[30–37], and it is well established that quaternary organo-ammonium salts show diverse reactivity and synthetic utility as efficient substitutes for halides. However, the Stille reaction via C–N bond cleavage in such substrates has remained a missing piece in cross-coupling chemistry.

Herein, we describe the first Ni-catalysed Stille coupling reaction of C–N electrophiles (quaternary organo-ammonium salts). Although Ni catalysts are generally much less expensive than the corresponding Pd catalysts, they have been rarely used for Stille coupling[23]. We show that the reaction mediated by commercially available Ni-catalyst/imidazole ligand and CsF provides high synthetic efficiency and broad functional group compatibility.

## Results

### Ni-catalysed couplings of ammonium salts and stannanes.
Aryltrimethylammonium salts **1** were readily synthesized from various anilines/amines bearing $NH_2$, NHMe or $NMe_2$ groups (Fig. 1). We started to examine the reaction of **1** with aryltrimethylstannane **2**, which can be readily obtained by utilizing our recently developed naphthalene-catalysed quantitative synthesis of stannyl lithium[38], or by employing functionalized Grignard reagents or other organometallics[39]. After the extensive experimentation (Supplementary Tables 1–3), we identified the optimum conditions as heating a mixture of **1** (as triflate) and **2** in 1:1 ratio in dioxane with CsF as base, Ni(cod)$_2$ as catalyst and 1,3-dicyclohexylimidazol-2-ylidene (ICy) as ligand. We next examined the scope of the reaction for various ammonium salts **1**, with **2a** as a representative stannane. The results are summarized in Table 1.

We found that (1) tolyltrimethylammonium triflates were converted very smoothly and selectively to the corresponding coupling products, and the location (ortho-, meta- or para-) of the methyl substituent did not greatly affect the reaction (**3aa-ca**), though a highly sterically demanding substrate slowed down the reaction (**3da**); (2) substrates bearing electron-donating groups are less reactive (**3aa-ea**), while those with electron-withdrawing groups, including polar functional groups such as fluorine (**3fa-ga**), ester (**3ha**), ketone (**3ia**), nitrile (**3ja**), silyl (**3ka**) and sulfone (**3la**), which are generally sensitive to bases and organometallics, showed excellent reactivity; (3) other aromatic substrates such as biphenyl or naphthalene also reacted efficiently (**3ma-na**), indicating that molecules with expanded π-conjugated systems might be prepared by employing this method. Next, the reactivities of various stannanes **2** were investigated. Except in a few cases, such as very bulky substrates (**3hb**), the reactions proceeded smoothly to afford biaryl products with broad functional group tolerance (**3hc-ii**). Heterocyclic substrates reacted without difficulty, and the desired products were obtained in high yields (**3hj-hm**).

Several additional reactions are noteworthy, and illustrate further synthetic applications of this method for selective preparations of functional molecules (Fig. 2). First, compound **3he** synthesized via the present coupling reaction could be easily transformed into the ammonium salt (**1o**), which underwent further coupling with a second stannane **2a** to generate the p-terphenyl derivative (**3oa**) (Fig. 2a). Second, we focused on the

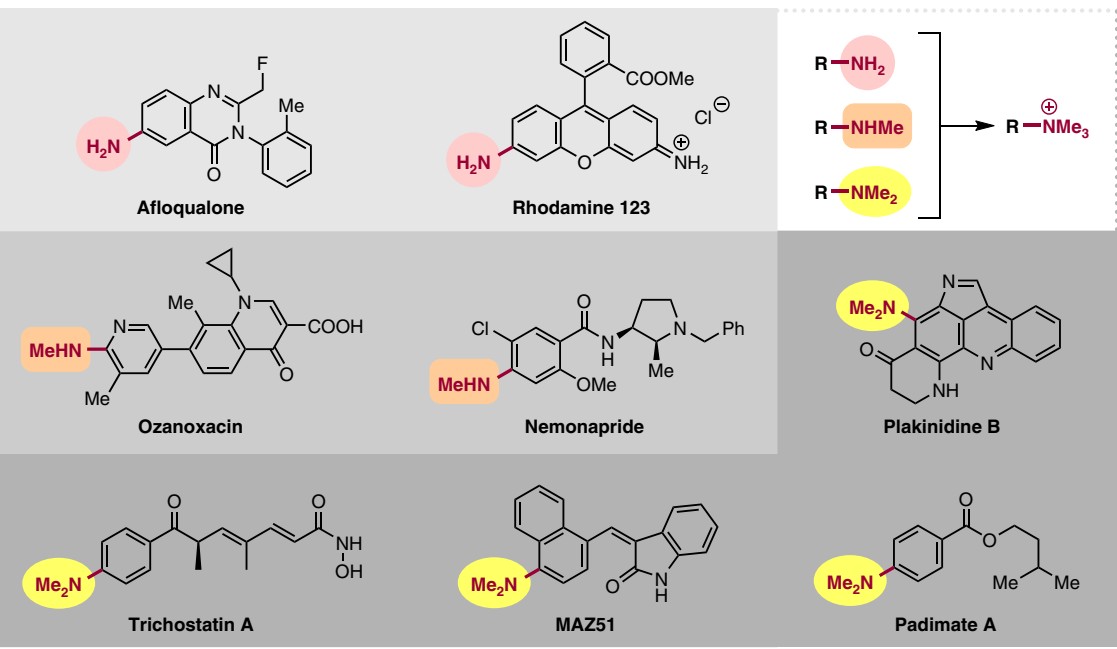

**Figure 1 | Amines.** Various functional molecules containing amine moieties.

**Table 1 | Stille cross-coupling reactions of aryltrimethylammonium salts 1 with aryltrimethylstannanes 2, leading to biaryl products 3, catalyzed by Ni(cod)₂ and ICy ligand.**

fact that NR$_2$ is often employed as a directing group in various aromatic reactions, such as Friedel–Crafts reactions and aromatic C–H functionalizations. For example, Ingleson[40] recently reported direct arene borylation (directed *p*-borylation) via electrophilic substitution of borenium. By combining this reaction with the current coupling reaction, *p*-terphenyl derivative (**3ma**) can also be synthesized from *N,N*-dimethylaniline via sequential reactions (Fig. 2b). These results clearly open up a new avenue for highly regio-controlled synthesis of multi-substituted arenes by utilizing amino groups on aromatic rings. Third, we have demonstrated that selective phenylation of an amino group can be achieved by using the ammonium salt of Padimate A, an ingredient in some sunscreens (Fig. 2c). In this reaction, the ester moiety was untouched, indicating the potential applicability of this method for late-stage derivatization of various functional molecules. Finally, benzyltrimethylammonium salt **4a** also reacted smoothly with stannane to give the coupling product **5aj** in excellent yield,

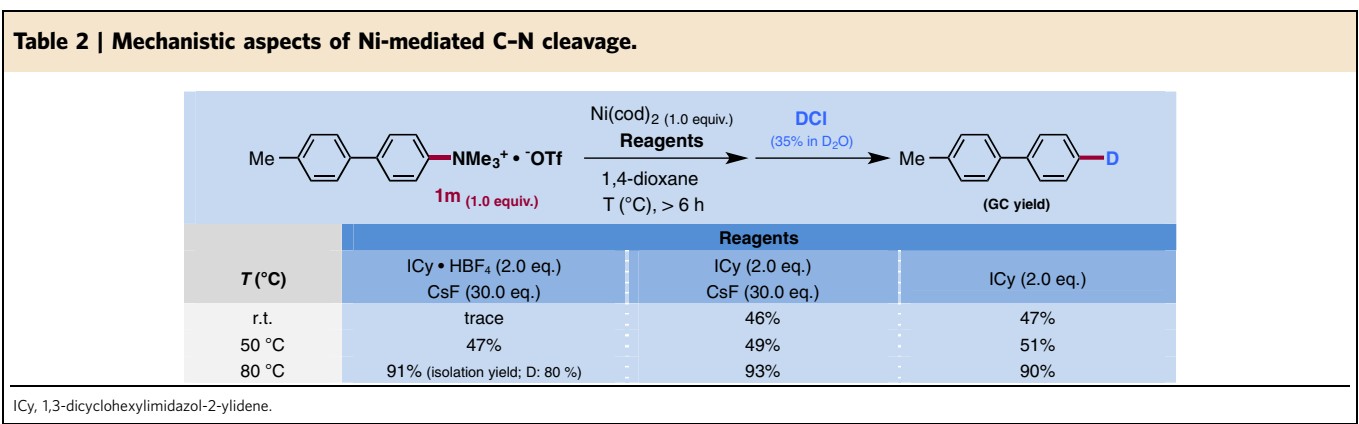

**Figure 2 | Synthetic applicability.** (**a,b**) Sequential cross-coupling for regio-controlled synthesis of *p*-terphenyl derivative; (**c**) Selective phenylation of NMe₂ group in Padimate A; (**d**) Cross-coupling between benzyltrimethylammonium salt **4a** and stannane **2j** catalyzed by Ni(cod)₂ and ICy ligand.

**Table 2 | Mechanistic aspects of Ni-mediated C–N cleavage.**

| T (°C) | ICy • HBF₄ (2.0 eq.) CsF (30.0 eq.) | ICy (2.0 eq.) CsF (30.0 eq.) | ICy (2.0 eq.) |
|---|---|---|---|
| r.t. | trace | 46% | 47% |
| 50 °C | 47% | 49% | 51% |
| 80 °C | 91% (isolation yield; D: 80 %) | 93% | 90% |

ICy, 1,3-dicyclohexylimidazol-2-ylidene.

suggesting broad applicability of this method to compounds containing a C(sp³)–N bond[19] (Fig. 2d).

**Mechanistic study**. The putative mechanism of our reaction involves (A) Ni(0)-mediated C–N bond cleavage of ammonium salt Ar–NMe₃⁺ to afford Ar¹–'Ni(II)' species (that is, an aromatic substitution reaction of NMe₃ by the electron-rich Ni⁰ complex), (B) transmetalation at the Ni(II) centre with Ar²SnMe₃ to afford Ar¹–'Ni(II)'–Ar² species, and finally (C) reductive elimination to give the final product with regeneration of Ni(0). While steps B and C are essentially the same as those of the conventional Stille reaction[41–48], step A involving C–N bond cleavage is of particular interest. Therefore, this step was investigated in detail in the stoichiometric reaction of ammonium triflate **1m** with Ni(cod)₂ and ICy ligand (Table 2). At room temperature, the reaction proceeded very sluggishly in the presence of ICy•HBF₄ and CsF (corresponding to the coupling conditions), and only a trace amount of the desired product was obtained. On the other hand, C–N cleavage was observed in the presence of neutral ICy ligand, with or without CsF.

Higher product yields were obtained in all cases at 50 °C, and the yield was further increased at 80 °C. These results imply that CsF acts simply as a base to release free ICy ligand in this step.

After several attempts, we obtained X-ray-grade crystals from a 1,4-dioxane solution of the stoichiometric reaction mixture containing Ni(cod)₂, ICy•HBF₄ and CsF in a ratio of 1:2:30 (Fig. 3a,b)[29]. The crystal structure is that of a prototype *trans*-Ni(ICy)₂Ar^(1m)F (Cambridge Crystallographic Data Centre (CCDC) 1438708) having a square-planar Ni(II) geometry, with two ICy ligands coordinated to the nickel atom in a *trans* arrangement. The compound was very stable, and even when it was treated with aqueous deuterium chloride (DCl) (0.2 mol l⁻¹) for several hours, the C–Ni bond remained unbroken; F/Cl exchange took place instead to give *trans*-Ni(ICy)₂Ar^(1m)Cl (confirmed by X-ray diffraction analysis, CCDC 1438709; Fig. 3a,c). Furthermore, the reaction of *trans*-Ni(ICy)₂Ar^(1m)F with PhSnMe₃ under the same conditions was very reluctant, indicating that this product may be a dead-end species, rather than the active intermediate of the present coupling reaction (Fig. 3a). Some important conclusions regarding the reaction

**Figure 3 | Mechanistic study.** (**a**) Trapping of Ni-intermediates after C–N cleavage. (**b**, **c**) Crystal structures images.

**Table 3 | Influence of fluoride on the present coupling reaction.**

| $T$ (°C) | Reagents | |
|---|---|---|
| | ICy • HBF$_4$ (20 mol%) CsF (3.0 eq.) | ICy (20 mol%) |
| r.t. | N.D. | N.D. |
| 50 °C | 45% | N.D. |
| 80 °C | 95% | N.D. |

N.D., not determined; r.t., room temperature.

mechanism of this coupling can be drawn from the data in Table 2 and Fig. 3, that is, (1) when a substantial amount of PhSnMe$_3$ is present in the C–N bond cleavage step, the subsequent transmetalation proceeds smoothly, but (2) in the absence of PhSnMe$_3$, the reaction affords a dead-end species, that is, *trans*-Ni(ICy)$_2$ArF, which is inert to transmetalation with PhSnMe$_3$.

The role of fluoride ion and the nature of the post C–N bond cleavage step in the present coupling reaction were investigated in model systems (Table 3). Several elegant investigations to clarify the influence of fluoride in Stille coupling reactions have already been reported[41–44]. In our case, when salt-free neutral ICy was used in the absence of CsF, the coupling reaction did not proceed at all. On the other hand, under the standard conditions, the reaction temperature played a crucial role in determining the coupling yield. The coupling did not occur at room temperature, but proceeded at higher temperature, and was accelerated as the temperature was increased. The yield of product **3ha** reached 95% at 80 °C. These results suggest that fluoride is necessary for Ni/Sn transmetalation, which may be quite energy-demanding.

**DFT calculations.** Next, we employed density functional theory (DFT) calculations at the B3LYP (refs 49–51)/M06 (ref. 52) level,

together with the artificial force-induced reaction method[53,54], to examine in detail the mechanism of this cross-coupling reaction. The results are illustrated in Fig. 4. First, the Ni(0) − π complex **CP0** is formed with − 3.0 kcal mol$^{-1}$ exothermicity from Ni(ICy)$_2$ (generated from Ni[cod]$_2$ and ICy) and [PhNMe$_3$]$^+$ F$^-$ (generated via anion metathesis of [PhNMe$_3$]$^+$ [OTf]$^-$ and CsF; the reaction route starting from [PhNMe$_3$]$^+$ [OTf]$^-$ was also calculated, but there was no marked difference in geometric structure or energy profile, compared with the results shown in Fig. 4). From **CP0**, Ni(0) can migrate on the phenyl ring to the proximal position of the C–N bond via **TS0** with an energy loss of only 10.2 kcal mol$^{-1}$ to form the more stable **CP1**. Cleavage of the C–N bond then takes place very smoothly as a S$_N$Ar process (**TS1**, − 2.0 kcal mol$^{-1}$), with release of NMe$_3$, affording intermediate **CP2-1** with large exothermicity (− 45.5 kcal mol$^{-1}$). The two ICys in **CP2-1** arrange in the *cis*-position, in which the horizontal Ni–C$^{(ICy)}$ bond ($d^2 = 2.01$ Å) is rather longer than the vertical one ($d^2 = 1.92$ Å). PhSnMe$_3$ then approaches the Ni(II) centre in **CP2-1** after the loss of one ICy ligand and rotation of the Ni–F bond from the vertical to the horizontal position (Supplementary Fig. 1) to generate **CP2-2** with an overall energy loss of 18.4 kcal mol$^{-1}$. To reach the TS of transmetalation, **TS2**, the orientation of the phenyl group of PhSnMe$_3$ changes so that the sp2-orbital bound to the Sn metal can interact with the Ni(II) centre, and the C–Sn bond is cleaved with a small activation energy (4.1 kcal mol$^{-1}$) to give **CP3-1**

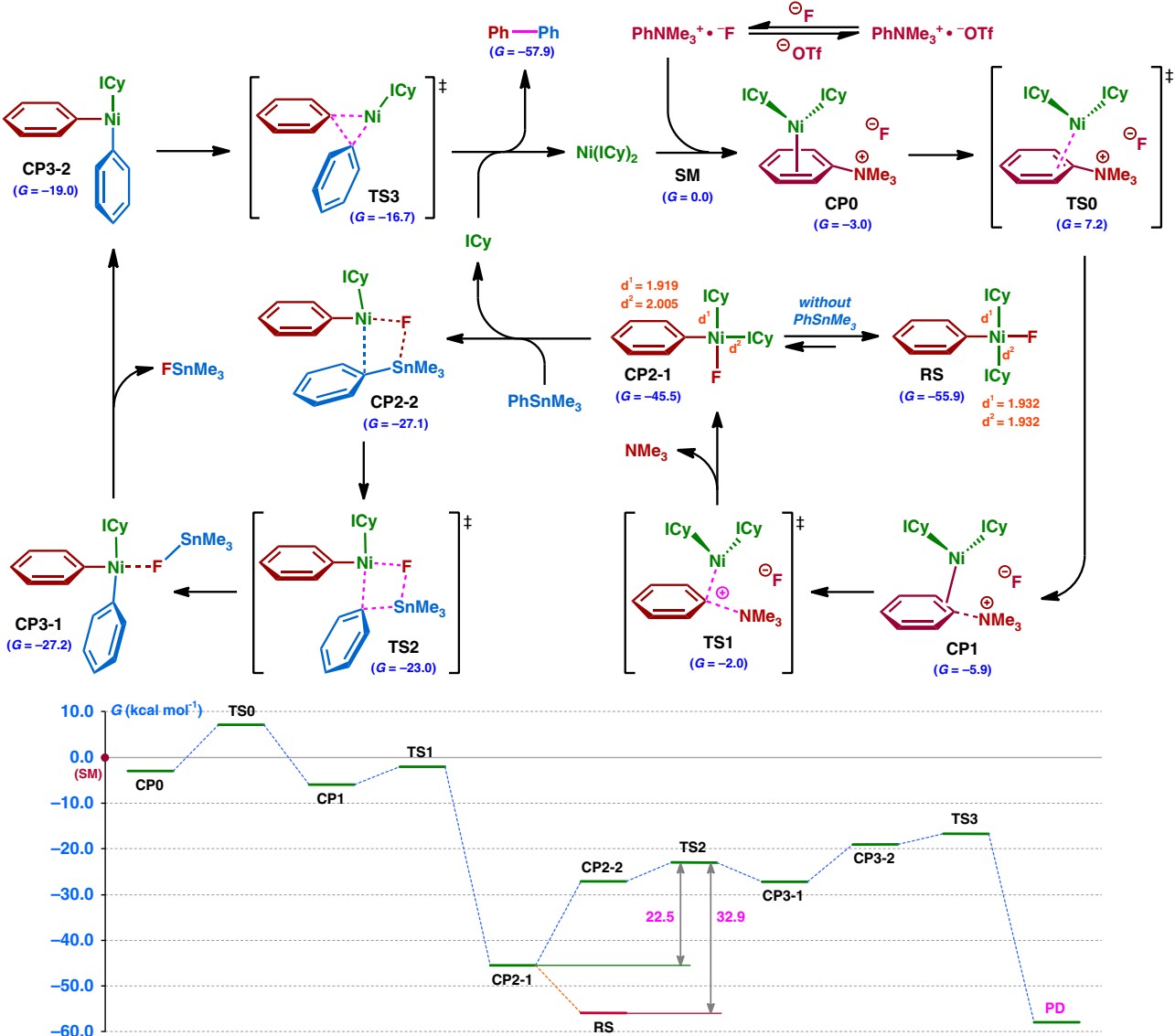

**Figure 4 | Reaction route located by means of DFT calculations.** Optimizations: B3LYP/LANL2DZ (Ni, Sn)&6-31G* (H, C, N and F). Single-point energies: M06(polarizable continuum model, solvent = 1, 4-dioxane)/SDD (Ni, Sn)&6-311 + + G** (H, C, N and F).

( − 27.2 kcal mol⁻¹). **CP3-1** then ejects FSnMe₃ to afford the precursor for the reductive elimination, **CP3-2** ( − 19.0 kcal mol⁻¹). Finally, C–C bond formation proceeds smoothly through **TS3** with an energy loss of only 2.3 kcal mol⁻¹ to produce the final product, Ph–Ph, and the Ni(ICy)₂ catalyst is regenerated with a large energy gain. We also carried out the experimental and theoretical studies of the possible alternative Ni(I)/Ni(III) pathway (Supplementary Figs 2–4; Supplementary Discussion). Although we cannot completely rule out the involvement of the Ni(I)/Ni(III) mechanism, and other scenarios could be contemplated, the computational and experimental results are all consistent with the view that the Ni(0)/Ni(II) route is more favourable and would be at least the predominant reaction pathway.

The computed resting-state **RS** (*trans*-isomer of **CP2-1**) is essentially identical with the structure obtained by X-ray diffraction analysis (Fig. 3b). **RS** is thermodynamically more favourable than **CP2-1** by 10.4 kcal mol⁻¹, and thus the total energy loss for the transmetalation (**TS2**) from **RS** (32.9 kcal mol⁻¹) is much higher than that from **CP2-1**

(22.5 kcal mol⁻¹), suggesting low reactivity (Fig. 4), which is in good agreement with the experimental facts.

We next examined the role of fluoride ion in the transmetalation step (Fig. 5)[41–44]. In the presence of fluoride ligand on the Ni(II) metal, this step is facilitated by a push-pull interaction, where F anion plays the role of a Lewis basic activator coordinating to the Sn metal to enhance the transfer ability of the Ph group of PhSnMe₃, and the Ni(II) centre is activated by the F•••SnMe₃ interaction, allowing the Ph group to undergo smooth bond switching from Sn to Ni(II), affording the biarylated Ni(II) intermediate **CP3-1** with an activation energy of only 4.1 kcal mol⁻¹. On the other hand, **CP2-2$^{OTf}$** with triflate anion on the Ni(II) undergoes transmetalation via a six-membered TS **TS2$^{OTf}$** in essentially the same manner. However, transmetalation of **CP2-2$^{OTf}$** requires much higher activation energy (19.5 kcal mol⁻¹) than that of **CP2-2**. In addition, the resultant intermediate **CP3-1$^{OTf}$** is geometrically and electronically unstable, and is thermodynamically disfavoured by 15.0 kcal mol⁻¹ compared with **CP2-2$^{OTf}$**. Thus, transmetalation without the assistance of fluoride anion should encounter both kinetic and

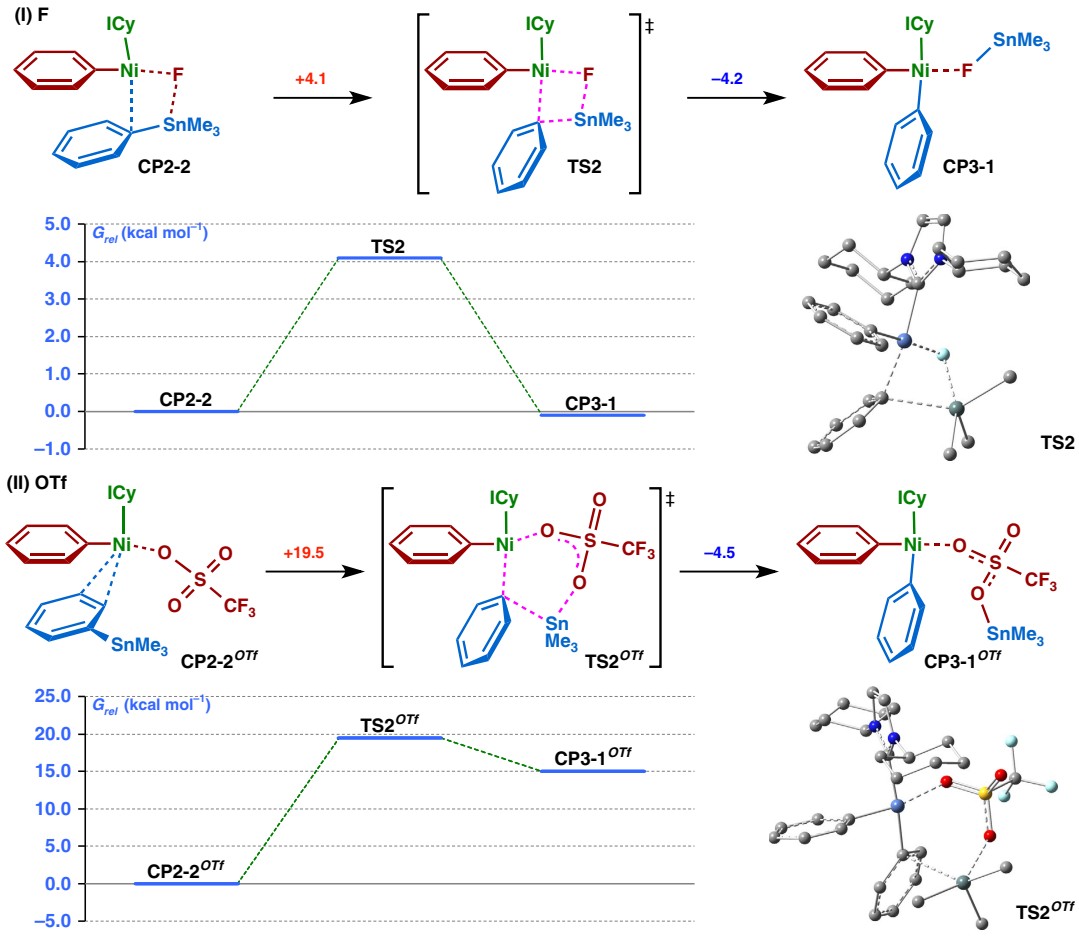

**Figure 5 | Transmetalation step.** DFT calculation for the transmetalation step in the presence and absence of fluoride. See Fig. 4 for details.

thermodynamic difficulties, in accordance with the experimental facts.

## Discussion

Almost 40 years after the discovery of Stille coupling chemistry, we have developed the first Stille reaction of quaternary ammonium salts via C–N bond cleavage catalysed by a commercially available Ni-catalyst/ligand. Few Ni-catalysed Stille coupling reactions have been reported so far. This novel C–N type Stille reaction is expected to be a powerful tool for the straightforward transformation of various aromatic amines/quaternary ammonium salts into multi-aromatic compounds and oligo(arylene)s, which have many potential applications in pharmaceutical, agrochemical and materials sciences. We also provide the first comprehensive reaction profile of cross-coupling via C–N bond cleavage, uncovered by employing a combination of experimental and computational methods. Work to extend the scope of this reaction and to apply it for synthesis of a range of functional molecules, including extended $\pi$-conjugation systems and bioactive compounds, is in progress.

## Methods

**General methods.** All reactions were carried out under a slightly positive pressure of dry argon by using standard Schlenk line techniques or in a glovebox (Braun, Labmaster SP). The oxygen and moisture concentrations in the glovebox atmosphere were monitored with an $O_2/H_2O$ analyser to ensure both were always $<0.1$ p.p.m. Unless otherwise noted, all starting materials including dehydrated solvents were purchased from WAKO, KANTO, TCI or ALDRICH. Ammonium salts were prepared via reported protocols (Supplementary Methods). $ArSnMe_3$ were prepared through the reaction of (1) $Me_3SnCl$ with corresponding aryl lithium or (2) $Me_3SnLi$ with corresponding aryl bromides/iodides. Nuclear

magnetic resonance (NMR) spectra were obtained on JEOL AL-300, AL-400 NMR and/or BRUKER AVANCE III HD spectrometers. Column chromatography was performed with silica gel 60 (230–400 mesh) from Merck and thin-layer chromatography was carried out on 0.25 mm Merck silica gel plates (60F-254).

**Typical procedure for cross-coupling between 1 and 2.** A Schlenk tube was charged with an aryltrimethylammonium triflate **1** (0.5 mmol), $Ni(cod)_2$ (13.8 mg, 0.05 mmol), $ICy \cdot HBF_4$ (32.0 mg, 0.1 mmol), CsF (227.9 mg, 1.5 mmol), $ArSnMe_3$ **2** (0.55 mmol) and dioxane (5 ml) under an argon atmosphere. The reaction mixture was stirred at 80 °C overnight and then cooled to room temperature. Water (10 ml) was added and the resulting mixture was extracted with ethyl acetate (3 × 10 ml). The combined organic layer was dried over $Na_2SO_4$, filtered and concentrated. The residue was purified by column chromatography on silica gel to afford product **3**. Representative example: **3aa**, colourless oil, isolated yield 60%; $^1$H NMR (300 MHz, $CDCl_3$) $\delta$ 7.72–7.67 (m, 2H), 7.63–7.58 (m, 2H), 7.56–7.50 (m, 2H), 7.48–7.34 (m, 3H), 2.50 (s, 3H). $^{13}$C NMR (75 MHz, $CDCl_3$) $\delta$ 141.25, 138.45, 137.11, 129.55, 128.82, 128.78, 127.32, 127.24, 127.07, 127.04 and 21.00. For full experimental details, see Supplementary Figs 5–68 and Supplementary Methods. For details on synthesis, characterization and deuteration of $trans$-Ni$(ICy)_2$Ar$^{(1m)}$X (X = F and Cl), see Supplementary Figs 69–74 and Supplementary Methods.

**X-ray crystallographic analysis.** Data collections were performed at 100 K on a Bruker D8 VENTURE diffractometer (PHOTON-100 CMOS detector, IμS-microsource, focusing mirrors, CuKα $\lambda = 1.54178$ Å) and processed using Bruker APEX-II software. The structure was solved by SHELXT[55] and refined by full-matrix least squares on $F^2$ for all data using SHELXL[56] and ShelXle[57] software. All non-disordered non-hydrogen atoms were refined anisotropically, and hydrogen atoms except those of water molecules were placed in the calculated positions. In the refinement of $trans$-Ni$(ICy)_2$Ar$^{(1m)}$Cl, the disordered cyclohexane rings were refined with equivalent anisotropic displacement parameters (EADP). DELU restraints were applied to solvent $n$-pentane, which rides on a symmetrically special position. The hydrogen atoms of solvent water were located on the difference Fourier maps and refined isotropically. DFIX and DANG restraints were applied to most of these hydrogen atoms (Supplementary Fig. 75; Supplementary Tables 4 and 5).

**Computation methods.** All DFT calculations were performed with the Gaussian 09 program (Revision D.01)[58] and the GRRM 11 (Version 11.03 (ref. 59) based on Gaussian 09) program. Structure optimizations were carried out at the B3LYP level in the gas phase, using the LANL2DZ basis set[60–62] for Ni and Sn, and the 6-31G* basis set[63,64] for H, C, N and F (keyword five dimensional was used in the calculations). The vibrational frequencies were computed at the same level to check whether each optimized structure is an energy minimum (no imaginary frequency) or a transition state (one imaginary frequency), and to evaluate its zero-point vibrational energy and thermal corrections at 298 K. Intrinsic reaction coordinates were calculated to confirm the connection between the transition states and the reactants/products. The single-point energy considering the solvent effect of 1,4-dioxane was obtained via calculation of the B3LYP geometries with M06 functional theory, using the Stuttgart-Dresden ECP (effective core potentials) and D-basis set (SDD) basis set[65,66] for Ni and Sn, and the 6-311++G** basis set[67,68] for other atoms. Solvation was evaluated by the self-consistent reaction field method using the polarizable continuum model[69]. The Gibbs free energy used for the discussion in this study was calculated by adding the gas-phase Gibbs free energy correction and the solution-phase single-point energy. Geometry of ICy was taken from the crystal structures. Original energy profiles and cartesian coordinates for DFT calculation be found in the Supplementary Data 1.

**Data availability.** Detailed experimental procedures and characterization of compounds can be found in the Supplementary Figs 5–75; Supplementary Methods. Original energy profiles and cartesian coordinates for DFT calculation be found in the Supplementary Data 1. Crystallographic data have been deposited with the CCDC, with deposition number CCDC-1438708 (*trans*-Ni(I-Cy)$_2$Ar$^{(1m)}$Cl) and 1438709 (*trans*-Ni(ICy)Ar$^{(1m)}$Cl). These data can be obtained free of charge from the CCDC via www.ccdc.cam.ac.uk/data_request/cif. All other data are available from the authors on reasonable request.

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

### Acknowledgements

This work was supported by JSPS KAKENHI (S) (no. 24229011; to M.U.), JSPS Grant WAKATE-B (no. 24790035; to C.W.), and JSPS TEI-SOKU Program (PU14008; to C.W.). This research was also partly supported by grants (to M.U.) from Takeda Science Foundation, Asahi Glass Foundation, Daiichi-Sankyo Foundation of Life Sciences, Mochida Memorial Foundation, Tokyo Biochemical Research Foundation, Nagase Science Technology Development Foundation, Yamada Science Foundation and Sumitomo Foundation. D.Y.W. is grateful for a scholarship provided by the Uehara Memorial Foundation.

### Author contributions

D.Y.W. planned, conducted, analysed and summarized the experiments. M.K., S.K. and K.Y. performed X-ray diffraction analysis. C.W. conceived the project and performed DFT calculation. Z.K.Y. and K.M. participate in the experiments or discussions. D.Y.W., C.W. and M.U. wrote the manuscript with feedback from all authors. C.W. and M.U. supervised the research.

### Additional information

**Competing financial interests:** The authors declare no competing financial interests.

