## [Peer Review File · Nature Communications]

Reviewers' comments:

Reviewer #1 (Remarks to the Author):

The submitted work describes a Ni-catalyzed Stille reaction between aryltrimethylammonium electrophiles and ArSnMe_3 nucleophiles. Since there are almost no examples of nickel-catalyzed Stille reactions, a synthetic and mechanistic study of such a process could be highly valuable if mechanistic insight or novel methodology results. As the authors correctly describe in the second paragraph of this manuscript, the use of organic trimethylammonium salts in cross-coupling reactions has much precedent in Ni- and Pd-catalyzed systems. Therefore, the work presented in this manuscript must be judged by the novelty of the chemistry that is enabled, as opposed to the fact that trimethylammonium leaving groups are employed. While synthetic details of this manuscript are solid, the overall utility of the described process does fall short of what I would expect for a manuscript published in Nature Comm. All products can be readily prepared by other cross-coupling methods. Thus, the immediate utility of the described process is not obvious. The iterative cross-coupling reaction shown in Figure 1A is very nice work, and is the most novel synthetic aspect of this chemistry.

Larger, fundamental problems stem from the mechanistic work described in the manuscript. The conclusions made from the studies involving the stoichiometric compounds in Figure 2 are not convincing. The authors show that the $\text{NHC}_2\text{Ni(II)ArF}$ complex is not competent in stoichiometric studies with organostannane nucleophiles. However, they propose that "when a substantial amount of PhSnMe_3 is present in the C-N bond-cleavage step, the subsequent transmetalation proceeds smoothly." From the data presented in Figure 2, I do not believe that this conclusion can possibly be made. Their proposal is later corroborated by DFT studies that show an unfavorable isomerization to $\text{trans-NHC}_2\text{Ni(II)ArF}$. However, these data also open the possibility of a different catalytic cycle. In this study, the authors are only considering a Ni(0)-Ni(II) redox cycle. However, work from the groups of Louis (Organometallics 2011, 30, 2546) and Matsubara (Chem. Commun. 2010, 46, 1932) have implicated the possibility of a Ni(I)-Ni(III) redox cycle for analogous processes using NHC ligands. I do not feel that we can overlook the possibility of such a mechanism here - particularly since Ni(0) and Ni(II) complexes are well-known to disproportionate into Ni(I) complexes. If the DFT analysis took the Ni(I) pathway into consideration, then the proposed Ni(0)-Ni(II) pathway would be more convincing. I feel that analysis of this alternate pathway must be performed before much credence can be placed in the mechanism proposed by the authors, and before this manuscript can be considered for publication in any journal.

The journal title and names of authors are missing from reference 10.

Reviewer #2 (Remarks to the Author):

A. Summary of the key results

Prof. Uchiyama performed Stille type cross-coupling between arylammonium and organotin compounds. Although the various cross coupling reactions starting from arylammonium salt were already reported, a use of organotin was not reported. Not only aryltin but also arylmethyltin was also examined. The detailed mechanistic study was also performed and imply the RD is the transmetalation step. In this step, F ion plays a crucial role.

B. Originality and interest: if not novel, please give references

The combination of ammonium salt and organotin is novel.

C. Data & methodology: validity of approach, quality of data, quality of presentation

D. Appropriate use of statistics and treatment of uncertainties

E. Conclusions: robustness, validity, reliability

No problem for the above three questions.

F. Suggested improvements: experiments, data for possible revision

I want to suggest that the geometry around tin atom is better to be shown in Table 5.

G. References: appropriate credit to previous work?

H. Clarity and context: lucidity of abstract/summary, appropriateness of abstract, introduction and conclusions

No problem for the above two questions.

Neutral amino group is much preferable, as the original Stille coupling is classified as a reaction under the neutral condition, compare to the other cross-coupling. Use of ammonium and cesium fluoride might be claimed. In my opinion, however, this is the first example. So it may be appear in this urnal.

Reviewer #3 (Remarks to the Author):

I was asked to review the crystallographic work presented. I am not qualified to comment on the impact of the synthetic work in this ms - and so I concentrate my remarks entirely on the quality of the crystal structures provided.

The structures in this paper appear to be new and they are used only to prove the chemical identity of the species described. For this purpose they are acceptable. (Note that other Ni(aryl)XL₂ structures with L = carbene are known. Thus the structures do not add any great impact in themselves to the paper).

Although the structures are fit for purpose, their final refinement and the reporting of them needs to be tweaked. If the paper is accepted by the other referees then the following points should be addressed before final acceptance.

1. Both structures have somewhat low % data completeness. Both cif files comment that this is due to weak diffraction intensity. These comments should be removed as (a) weak I and completeness have nothing to do with each other; and (b) the ratios of observed to unique reflections (7432/8116 and 12027/14372) indicate that the average I is in fact rather high. It is too late to sort out the low completeness now, but for future reference the authors should note that this looks very much as if their data collection strategy was not correct.

2. For the Cl structure the main reason for the poor quality is almost certainly twinning. The cif mentions that this is so but does not indicate if anything was done to model this. The authors should go back to the raw data and attempt to process the data as a TWIN (perhaps by 180 deg around 100 ??). They should also attempt a post-processing twin treatment using say ROTEX. If one of these twin treatments works then the authors should of course replace their current version of the structure with the new one. If no twin treatment is suitable then the authors should describe what they have done (and that it was unsuccessful) both in the cif and in the experimental section of the ESI.

3. Some of the poorest fitting reflections in the Cl structure are from very low angle reflections ("behind the beamstop"). The authors should consider removing these from the dataset.

4. For the F structure. The main paper text and the ESI differ on the solvent used to crystallise this.

5. For the F structure. The Cy ring of C60 is disordered. The authors should model it as such and re-refine. This should get rid of the worst Q peaks.

6. For the F structure. The O atoms for water look as though their positions are also disordered and/or not 100% occupied. The authors should attempt to model this.
7. The missing H atoms on water. These positions should be calculated by determining suitable H-bonding geometries.
8. For the F structure. Z should = 4. This will correct the formula given.
9. In the cif and/or the ESI. An explanation should be given for the DELU restraints used.
10. In the ESI both independent molecules of the F complex should be drawn.

Reviewer #4 (Remarks to the Author):

In the present work the authors developed a Ni catalyzed Stille coupling between quaternary ammonium salts and arylstannanes. As the authors state, this is a missing piece in cross-coupling chemistry. Moreover, the use of non-precious metals as catalysts (even for already developed reactions) is one of the goals in homogeneous catalysis applied to synthetic methods. Thus, in principle, the topic is suitable for publication in Nature Chemistry.

The reaction mechanism has been analyzed by means of DFT calculations at the B3LYP/M06 level (a standard level of theory). In order to analyze the potential energy surface the authors used the AFIR method developed by Maeda and Morokuma. The proposed reaction mechanism is relatively similar to the "cyclic" mechanism commonly accepted for the Pd-catalyzed Stille cross-coupling. The first step, however, is not an oxidative addition, it is better described as an aromatic substitution (S_NAr) where the NMe₃ is substituted by the Ni complex; the energy barrier for this step is rather low. The barrier for this process is not affected by having F⁻ or OTf⁻ in the calculations (I assume forming ion pairs).

Once intermediate CP2-1 is formed, next step is substitution of Icy ligand by stannane (along with a cis-to-trans isomerization). This step is found to be endothermic but the energy barrier is not calculated. I do not think this is going to be the rate determining step, but authors should try to calculate the barrier associated to this step, or at least comment on how it may take place, in case it has been described for other related Ni complexes in the literature.

The major effect of F⁻ is, as expected, in the transmetalation step. The barrier for the transmetalation step is very different when using F⁻ (~ 4 kcal/mol) or OTf⁻ (~20 kcal/mol). The F⁻ is acting as Lewis base for coordinating the stannane and facilitating the transmetalation. As far as the reaction conditions is concerned, it is noteworthy that the ratio among the catalyst:ligand:CsF is 1:2:30. The number of equivalents of CsF is quite large. Do the authors have any clue about the reason for this? The necessity of using such a large concentration is something not reflected in the mechanistic proposal. The authors should try to find an explanation for this. Finally, the rate determining step is the reductive elimination step, as in many cases in the related Pd-catalyzed reaction.

Overall in my opinion the work may deserve to be published in Nat. Commun. but the authors should first address the points previously commented.

In addition, references need to be revised: some of them are not properly numbered. For instance, there are 50 references in the text, whereas there are 49 in the reference section.

A very recent review dedicated to Transition-Metal-Catalyzed Cleavage of C-N Single Bonds (Chem. Rev., 2015, 115 (21), pp 12045-12090) might be added.

Revision Details

Reviewer 1:

[1] The submitted work describes a Ni-catalyzed Stille reaction between aryltrimethylammonium electrophiles and ArylSnMe₃ nucleophiles. Since there are almost no examples of nickel-catalyzed Stille reactions, a synthetic and mechanistic study of such a process could be highly valuable if mechanistic insight or novel methodology results. As the authors correctly describe in the second paragraph of this manuscript, the use of organic trimethylammonium salts in cross-coupling reactions has much precedent in Ni- and Pd-catalyzed systems. Therefore, the work presented in this manuscript must be judged by the novelty of the chemistry that is enabled, as opposed to the fact that trimethylammonium leaving groups are employed. While synthetic details of this manuscript are solid, the overall utility of the described process does fall short of what I would expect for a manuscript published in Nature Comm. All products can be readily prepared by other cross-coupling methods. Thus, the immediate utility of the described process is not obvious. The iterative cross-coupling reaction shown in Figure 1A is very nice work, and is the most novel synthetic aspect of this chemistry.

Answer: We thank the reviewer for the comments on the novelty and importance of our work. To strengthen this aspect and further demonstrate the utility of this chemistry, we have added more content and inserted Figure 1 in the Introduction part, and we have also added two more examples of synthetic applications (Scheme 1 and Fig. 2 in the revised MS). Firstly, combining the Friedel–Crafts type arene borylation and the current coupling reaction leads to an efficient and selective synthesis of terphenyl derivative, indicating that an amino group on an aromatic ring can be utilized as a toehold for regio-controlled synthetic strategies of multi-functionalized aromatics. Selective arylation of the NMe₂ group in Padimate A, a sunscreen component used to prevent direct DNA damage, was also demonstrated. We think this transformation would be difficult to achieve with traditional cross-coupling procedures.

Scheme 1

[2] Larger, fundamental problems stem from the mechanistic work described in the manuscript. The conclusions made from the studies involving the stoichiometric compounds in Figure 2 are not convincing. The authors show that the NHC₂Ni(II)ArF complex is not competent in stoichiometric studies with organostannane nucleophiles. However, they propose that "when a substantial amount of PhSnMe₃ is present in the C–N bond-cleavage step, the subsequent transmetalation proceeds smoothly." From the data presented in Figure 2, I do not believe that this conclusion can possibly be made. Their proposal is later corroborated by DFT studies that show an unfavorable isomerization to trans-NHC₂Ni(II)ArF. However, these data also open the possibility of a different catalytic cycle. In this study, the authors are only considering a Ni(0)–Ni(II) redox cycle. However, work from the groups of Louis (Organometallics 2011, 30, 2546) and Matsubara (Chem. Commun. 2010, 46, 1932) have implicated the possibility of a Ni(I)–Ni(III) redox cycle for analogous processes using NHC ligands. I do not feel that we can overlook the possibility of such a mechanism here - particularly since Ni(0) and Ni(II) complexes are well-known to disproportionate into Ni(I) complexes. If the DFT analysis took the Ni(I) pathway into

consideration, then the proposed Ni(0)-Ni(II) pathway would be more convincing. I feel that analysis of this alternate pathway must be performed before much credence can be placed in the mechanism proposed by the authors, and before this manuscript can be considered for publication in any journal.

Answer: We thank the reviewer for this comment, and agree that the original MS requires some additional experiments/computations from a mechanistic point of view. As for the steps involving the $\text{NHC}_2\text{Ni(II)ArF}$ intermediate, a more detailed discussion is included in the SI, as shown below (Scheme 2). After the formation of *cis*- $\text{NHC}_2\text{Ni(II)ArF}$ (**CP2-1**), one ICy ligand opposite the Ph group would leave to form **INT-a**. An in-depth scan of the potential energy surface showed that the energy change for the release of ICy is a simple uphill process (no TS). The C–F bond in **INT-a** then rotates from the *cis*- to the *trans*-position to Ph (**INT-b**), via a very low energy barrier (**TS-a_b**). PhSnMe_3 then approaches **INT-b** to form **CP2-2**, leading to a thermodynamically very stable coupling product. In the absence of PhSnMe_3 , as indicated by the control experiments in Fig 3, disassociated ICy would again coordinate to form *trans*-**RS**, which is ca. 10 kcal/mol more stable than **CP2-1**. Hence, the total activation barrier of the transmetalation from **RS** to **CP2-2** adds up to over 30 kcal/mol, which would be kinetically difficult under the current reaction conditions, as reflected in the very low yield in the control experiments (Fig 3). A similar process for Pd-catalysis has been reported in the literature (ref. 44).

Scheme 2

Secondly, regarding the Ni(I)-Ni(III) mechanism, we have performed several control experiments. When we used the Ni(I) catalyst (synthesized as reported in the literature cited by the referee) instead of the Ni(0) catalyst, the reaction became very sluggish, as shown in Scheme 3. Although the possibility of the Ni(I)-Ni(III) pathway could not be totally ruled out, the results of our experimental and computational investigations all indicate that the Ni(0)-Ni(II) pathway is more probable.

Scheme 3

Reviewer 2:

[1] I want to suggest that the geometry around tin atom is better to be shown in Table 5.

Answer: We have added the 3D-structure for the two TSs to show clearly the geometry around the tin atom in Figure 6 of revised MS.

Reviewer 3:

[1] Both structures have somewhat low % data completeness. Both cif files comment that this is due to weak diffraction intensity. These comments should be removed as (a) weak *I* and completeness have nothing to do with each other; and (b) the ratios of observed to unique reflections (7432/8116 and 12027/14372) indicate that the average *I* is in fact rather high.

It is too late to sort out the low completeness now, but for future reference the authors should note that this looks very much as if their data collection strategy was not correct.

Answer: We agree the low completeness of both structures is due to our incomplete dataset. We could not perform further diffraction experiments because have no more suitable single crystals of these compounds at present. Nevertheless, we think that both structures are adequate for the structure elucidation as the reviewer implied. However, we appreciate the reviewer's advice for future work. We revised our comment as follows.

Although the percent data completeness is a little low for these structures due to our incomplete datasets, the obtained structures are adequate for the present purpose.

[2] For the Cl structure the main reason for the poor quality is almost certainly twinning. The cif mentions that this is so but does not indicate if anything was done to model this. The authors should go back to the raw data and attempt to process the data as a TWIN (perhaps by 180 deg should go back to the raw data and attempt to process the data as a TWIN (perhaps by 180 deg around 100 ??). They should also attempt a post-processing twin treatment using say ROTEX. If one of these twin treatments works then the authors should of course replace their current version of the structure with the new one. If no twin treatment is suitable then the authors should describe what they have done (and that it was unsuccessful) both in the cif and in the experimental section of the ESI.

Answer: As the reviewer suggested, we carefully checked the raw data and tried to process the data as a TWIN (using Bruker cell_now and reciprocal viewer), but TWIN refinement did not work. We also attempted a post-processing twin treatment using TwinRotMat incorporating the PLATON program package, or ROTAX, but again, twin refinement was not successful. We revised the validation reply forms of the cif file of the Cl structure as follows.

We carefully checked the raw data and also tried to process the data as a TWIN, but this was unsuccessful. An attempt at post-processing twin refinement also failed.

[3] Some of the poorest fitting reflections in the Cl structure at from very low angle reflections ("behind the beamstop"). The authors should consider removing these from the dataset.

Answer: As the reviewer suggested, we removed some of the poorest fitting reflections in the lower angle reflections, and re-refined the structure by using 'OMIT'.

[4] For the F structure. The main paper text and the ESI differ on the solvent used to crystallise this.

Answer: We apologize for the inconsistency and have corrected the text and the ESI (pp S11-S12).

[5] For the F structure. The Cy ring of C60 is disordered. The authors should model it as such and re-refine. This should get rid of the worst Q peaks.

Answer: As the reviewer suggested, we tried to refine the Cy ring including C60 as disordered, but suitable structures were not obtained.

[6] For the F structure. The O atoms for water look as though their positions are also disordered and/or not 100% occupied. The authors should attempt to model this.

Answer: Again, we tried to refine the water O atoms as disordered, including the occupancy, but could not obtain stable structures.

[7] The missing H atoms on water. These positions should be calculated by determining suitable H-bonding geometries.

Answer: As mentioned above, we could not refine suitable water molecules, so we could not determine the water hydrogens.

[8] For the F structure. Z should = 4. This will correct the formula given.

Answer: We thank the reviewer for this comment. We set Z = 4 then re-refined the structure.

[9] In the cif and/or the ESI. An explanation should be given for the DELU restraints used.

Answer: According to the reviewer's comment, we revised the description of the use of the DELU restraints. In the `_refine_special_details` section of the cif file for the F structure, we changed the comment as follows: 'In the refinement of the solvent molecule (*n*-pentane C87, C88, C89, C90 and C91), which occupies a symmetrically special position, we applied 'DELU' restraints to C88, C89 and C90.'

[10] In the ESI both independent molecules of the F complex should be drawn.

Answer: We have done this as the reviewer suggested.

Reviewer 4:

[1] The reaction mechanism has been analyzed by means of DFT calculations at the B3LYP/M06 level (a standard level of theory). In order to analyze the potential energy surface the authors used the AFIR method developed by Maeda and Morokuma. The proposed reaction mechanism is relatively similar to the "cyclic" mechanism commonly accepted for the Pd-catalyzed Stille cross-coupling. The first step, however, is not an oxidative addition, it is better described as an aromatic substitution (S_NAr) where the NMe₃ is substituted by the

Ni complex; the energy barrier for this step is rather low. The barrier for this process is not affected by having F⁻ or OTf⁻ in the calculations (I assume forming ion pairs).

Answer: We agree and have corrected the corresponding illustration as suggested.

[2] Once intermediate CP2-1 is formed, next step is substitution of Icy ligand by stannane (along with a cis-to-trans isomerization. This step is found to be endothermic but the energy barrier is not calculated. I do not think this is going to be the rate determining step, but authors should try to calculate the barrier associated to this step, or at least comment on how it may take place, in case it has been described for other related Ni complexes in the literature.

Answer: Reviewer 1 also made a similar comment. Therefore, we made detailed calculations for the transformation step from CP2-1 to CP2-2 and have added them in SI and Scheme 2. We think these supplementary studies adequately clarify the situation.

[3] The major effect of F⁻ is, as expected, in the transmetallation step. The barrier for the transmetallation step is very different when using F⁻ (~ 4 kcal/mol) or OTf⁻ (~20 kcal/mol). The F⁻ is acting as Lewis base for coordinating the stannane and facilitating the transmetallation. As far as the reaction conditions is concerned, it is noteworthy that the ratio among the catalyst:ligand:CsF is 1:2:30. The number of equivalents of CsF is quite large. Do the authors have any clue about the reason for this? The necessity of using such a large concentration is something not reflected in the mechanistic proposal. The authors should try to find an explanation for this. Finally, the rate determining step is the reductive elimination step, as in many cases in the related Pd-catalyzed reaction.

Answer: CsF is a very common base for cross-coupling reactions, especially for Suzuki and Stille coupling, and it is normally used in excess amounts (over 2 equivalents, for example, see ref. 41-48). However, CsF is not readily soluble in common organic solvent such as ether, toluene, THF, or 1,4-dioxane, and hence the concentration of CsF is not high in the reaction solution.

[4] In addition, references need to be revised: some of them are not properly numbered. For instance, there are 50 references in the text, whereas there are 49 in the reference section. A very recent review dedicated to Transition-Metal-Catalyzed Cleavage of C-N Single Bonds (Chem. Rev., 2015, 115 (21), pp 12045-12090) might be added.

Answer: Thank you. We checked the reference numbers, and have also added the suggested review as ref. 11 in the original MS.

Reviewers' comments:

Reviewer #1 (Remarks to the Author):

In the revised manuscript submitted by Uchiyama et al., the authors have done little to assuage my initial concerns about the possibility of a Ni(I)-Ni(III) catalytic cycle being prevalent. A control reaction was performed using bis-ligated Ni(I)-NHC complexes bearing IMes and IPr as precatalysts despite the fact that IMes does not work as a supporting ligand, and IPr was not tested. [An aside: IPr should not be used to refer to the N-iPr-substituted NHC (as it is on S-2 of the supporting information) since this abbreviation is conventionally used to refer to the analogue of IMes with 2,6-di-isopropyl substitution.] The authors assume that catalytic activity obtained using 10 mol% of these precatalysts in the presence of 20 mol% Cy-NHC would be representative of a system where the Ni(I) complex bearing Cy-NHC was exclusively used. This is almost certainly not the case. I am actually surprised that 38% GC yield was obtained from this since 1/3 of the ligand present is not competent in the coupling reaction. The 38% yield nicely exemplifies the ability of a Ni catalyst to start as either Ni(0) or Ni(I) and still find its way to the required oxidation state to be active in Ni catalysis. This could be interpreted as evidence supporting Ni(I) catalysis. I apologize for being so difficult about this point, but it is essential that the Ni(I) pathway be convincingly ruled out for the computational work to be convincing. I appreciate that it is likely more complicated to conduct DFT calculations on Ni(I) systems, but I believe that a comparison is necessary. When this manuscript is published, it will be cited blindly by researchers as proof of an NHC-ligated Ni(0) cross-coupling mechanism. Without some calculations addressing the possibility of a Ni(I) pathway, the computational work is incomplete.

Other comment: Ref. 10 still does not have the names of the authors listed.

Reviewer #3 (Remarks to the Author):

Crystallographic referee report.

Most of the changes and/or replies that the authors have made in response to my original report are acceptable.

However, my points with respect to disorder in a cyclohexyl ring and in water solvent molecules have not been actioned properly. In the authors' reply (their points 5, 6 and 7) they claim that the disorder cannot be modelled and that H atom positions cannot be found. This is not so. I can model both ring and solvent disorder and this results in an improved and more accurate structure.

Whilst this is a minor point in relation to the authors' main chemical claims, I do not think that papers published in reputable journals should contain improperly refined data/models. This must be corrected before the paper is published.

Reviewer #4 (Remarks to the Author):

The revised version of the manuscript incorporates the suggestions I made in the previous report.

I must add that the concerns of referee 1 must be taken into consideration by the authors. Their control experiments suggest that reaction via Ni(I)-catalyst is not as clean as via Ni(0)-catalyst, but the reaction still works. Thus, as the authors state in their response, the Ni(I)-Ni(III) pathway

can not be ruled out according to their results.

I think this point is very important and the authors should have been more careful in the revised version. The authors, at least, must add these control experiments in the Sup. Inf. and make a proper comment in the main text concerning the mechanism.

Revision Details

Reviewer 1:

[1] I appreciate that it is likely more complicated to conduct DFT calculations on Ni(I) systems, but I believe that a comparison is necessary. When this manuscript is published, it will be cited blindly by researchers as proof of an NHC-ligated Ni(0) cross-coupling mechanism. Without some calculations addressing the possibility of a Ni(I) pathway, the computational work is incomplete.

Answer: We appreciate the reviewer's thoughtful comment on the Ni(I) pathway. In response, we have performed DFT calculation for the Ni(I)/Ni(III) mechanism at the same level as the Ni(0)/Ni(II) route (B3LYP&M06). The results are summarized in Scheme 1. Based on reported information, we envisioned that the Ni(I) catalyst $\text{Ni}^{\text{I}}(\text{ICy})_2\text{F}$ would be firstly generated from $\text{Ni}^0(\text{ICy})_2$ and $[\text{PhNMe}_3]^+\text{F}^-$, similar to the reported reaction between $\text{Ni}^0(\text{ICy})_2$ and ArX ($\text{X} = \text{Cl}, \text{Br}, \text{etc.}$). The resultant $\text{Ni}^{\text{I}}(\text{ICy})_2\text{F}$ reacts with PhSnMe_3 to form **CP-a**, $\text{Ni}^{\text{I}}(\text{ICy})_2\text{Ph}$, via Ni/Sn transmetalation with large endothermicity (+11.3 kcal/mol). Then, the Ni(I)- π complex **CP-b** is formed with a reasonable activation energy (+24.6 kcal mol⁻¹), albeit again with +13.3 kcal mol⁻¹ endothermicity. From **CP-b**, C-N bond cleavage takes place through **TS-b** with an energy loss of +6.0 kcal mol⁻¹. IRC analysis for **TS-b** failed to locate the proposed Ni(III) intermediate, and instead, a straightforward C-C bond formation occurs, leading directly to the final product.

Scheme 1. Reaction profile along the Ni(I) pathway

In summary, the theoretical calculations indicate that, as shown in Figure 1, the CPs and TSs in the Ni(0)/Ni(II) route (green line) are energetically more favorable than those in the Ni(I)/Ni(III) pathway (pink line), and the reaction is therefore much less likely to take place along the Ni(I)/Ni(III) route than along the Ni(0)/Ni(II) route.

Figure 1. Comparison of the energy profiles (ΔG , kcal/mol) of the Ni(0)/Ni(II) route (blue/green line) and the Ni(I)/Ni(III) route (red/purple line)

It is also important to note that in the stoichiometric reaction between Ni(0) catalyst and ammonium salts (Table 2 and Figure 3), no Ni(I) species was detected at all. Hence, although the possibility of the Ni(I)/Ni(III) mechanism cannot be totally ruled out, all the current computational and experimental results support the view that the Ni(0)/Ni(II) route is more favorable and would be at least the predominant reaction pathway. According to the comments of Referees 1 and 4, we have added the results of all these supplementary studies in the supporting information. This is also briefly mentioned in the text.

[2] Ref. 10 still does not have the names of the authors listed.

Answer: We apologize for the mistake and have corrected it.

Reviewer 3:

[1] However, my points with respect to disorder in a cyclohexyl ring and in water solvent molecules have not been actioned properly. In the authors' reply (their points 5, 6 and 7) they claim that the disorder cannot be modelled and that H atom positions cannot be found. This is not so. I can model both ring and solvent disorder and this results in an improved and more accurate structure.

Whilst this is a minor point in relation to the authors' main chemical claims, I do not think that papers published in reputable journals should contain improperly refined data/models. This must be corrected before the paper is published.

Answer: We gratefully appreciate the crystallographic referee's precise comments.

We again refined the structure (Fluorine complex) according to the referee's suggestion, and we succeeded in getting an improved structure. Both the cif and ORTEP drawing will be replaced.

The disordered cyclohexyl rings (occupancy 72/28) were treated with EADP constraints. In the refine_special_details section of the updated cif, the following description was added, "The disordered cyclohexane rings (C60, C60B, C61, C61B, C62, C62B, C63, C63B, C64, C64B, C65 and C65B) were refined with EADP constraints. Calculated occupancies of each cyclohexane ring were estimated to be 72:28."

The solvent water molecules were correctly assigned with hydrogen atoms included. Some hydrogen atoms were restrained with DFIX/DANG. In the refine_special_details section of the updated cif, the following description was added, "In the refinement of water molecules (O1, H1A, H1B, O2, H2A, H2B, O3, H3A and H3B), hydrogen atoms were located on the difference Fourier maps and refined isotropically. DFIX and/or DANG restraints were applied to all hydrogen atoms except H2A."

Reviewer 4:

[1] I must add that the concerns of referee 1 must be taken into consideration by the authors. Their control experiments suggest that reaction via Ni(I)-catalyst is not as clean as via Ni(0)-catalyst, but the reaction still

works. Thus, as the authors state in their response, the Ni(I)-Ni(III) pathway can not be ruled out according to their results.

I think this point is very important and the authors should have been more careful in the revised version. The authors, at least, must add these control experiments in the Sup. Inf. and make a proper comment in the main text concerning the mechanism.

Answer: We thank the referee for this important comment. As the referee pointed out, referee 1 also made a similar comment. Please refer to our response to referee 1 for full details of our response. Briefly, we made the suggested calculations for the Ni(I)/Ni(III) pathway and have added them in the SI together with the results of control experiments. The theoretical calculations indicate that the Ni(I)/Ni(III) route is far less favorable than the Ni(0)/Ni(II) route. In short, all the computational and experimental results support the view that the Ni(0)/Ni(II) route is more favorable and would be at least the predominant reaction pathway.

REVIEWERS' COMMENTS:

Reviewer #1 (Remarks to the Author):

The revised manuscript is certainly improved with the inclusion of at least cursory calculations of a Ni(I)-Ni(III) mechanism. However, there remain additional possible scenarios to consider for such a redox couple -- such as a bis-ligated, cationic complex preceding product-forming reductive elimination. While I am not entirely convinced by additional computation work, a thorough computational study may end up being a Herculean task for this Ni-catalyzed system, and would probably end up beyond the scope of the manuscript. Therefore, the manuscript should be publishable in its present form.

Reviewer #3 (Remarks to the Author):

The revised crystallographic work is now acceptable for publication.

Reviewer #4 (Remarks to the Author):

In the revised version of the manuscript the authors show by means of computational studies that the energy profile for the Ni(I)/Ni(III) mechanism is higher in energy. Thus, despite it can not be completely ruled out the Ni(0)/Ni(II) pathway looks much more feasible.

Accordingly, in my opinion, the manuscript can be accepted for publication.

AUTHOR RESPONSE TO REVIEWERS' COMMENTS:

[Reviewer 1] The revised manuscript is certainly improved with the inclusion of at least cursory calculations of a Ni(I)-Ni(III) mechanism. However, there remain additional possible scenarios to consider for such a redox couple -- such as a bis-ligated, cationic complex preceding product-forming reductive elimination. While I am not entirely convinced by additional computation work, a thorough computational study may end up being a Herculean task for this Ni-catalyzed system, and would probably end up beyond the scope of the manuscript. Therefore, the manuscript should be publishable in its present form.

[Reviewer 3] The revised crystallographic work is now acceptable for publication.

[Reviewer 4] In the revised version of the manuscript the authors show by means of computational studies that the energy profile for the Ni(I)/Ni(III) mechanism is higher in energy. Thus, despite it can not be completely ruled out the Ni(0)/Ni(II) pathway looks much more feasible.

Answer: We appreciate the reviewers' comments on the revised manuscript. In response to reviewers 1 and 3, and as suggested by the editor, we have added in the present manuscript a brief statement that alternative mechanisms cannot be completely ruled out.